# GBT: Generative Boosting Training Approach for Paraphrase Identification

**Rui Peng, Zhiling Jin, Yu Hong**[*]

School of Computer Science and Technology, Soochow University, SuZhou, China
{rpeng124, zhljinjackson, tianxianer}@gmail.com

## Abstract

Paraphrase Identification (PI), a task of determining whether a pair of sentences express the same meaning, is widely applied in Information Retrieval and Question Answering. Data Augmentation (DA) is proven effective in tackling PI task. However, the majority of DA methods still suffer from two limitations: inefficiency and poor quality. In this study, we propose the Generative Boosting Training (GBT) approach for PI. GBT designs a boosting learning method for single model based on the human learning process, utilizing seq2seq model to perform DA on misclassified instances periodically. We conduct experiments on the benchmark corpora QQP and LCQMC, towards both English and Chinese PI tasks. Experimental results show that our method yields significant improvements on a variety of Pre-trained Language Model (PLM) based baselines with good efficiency and effectiveness. It is noteworthy that a single BERT model (with a linear classifier) can outperform the state-of-the-art PI models with the boosting of GBT.

## 1 Introduction

PI can be boiled down to a binary classification task, aiming to determine whether pair of sentences convey the same or similar meaning. It is a fundamental natural language understanding task with non-trivial challenges, serving as a practical technique in the field of information retrieval and question answering (Hu et al., 2014; Cer et al., 2017; Rücklé et al., 2020; Pang et al., 2021).

A variety of neural PI approaches have been proposed (Chen et al., 2017; Wang et al., 2017; Zhang et al., 2017; Gong et al., 2017; Lai et al., 2019; Kim et al., 2019). Recently, the large PLM-based Models are leveraged as crucial supportive encoders for neural PI (Devlin et al., 2019; Liu et al., 2019; Cui et al., 2021; Zhang et al., 2021;

---

[*]Corresponding author.

| Sentences | Label | Edit Distance | Case type |
|---|---|---|---|
| 淘宝邮政编码怎么填? 
 (How to fill in the zip code of Taobao?) 
 淘宝邮政编码怎么改? 
 (How to modify the zip code of Taobao?) | 0 | 1 | $IBH_0$ |
| 管理者怎样才能让员工服从? 
 (How can managers get employees to obey?) 
 怎样使员工服从领导? 
 (How to make employees obey the leader?) | 1 | 8 | $IBH_1$ |

Table 1: $IBH$-type cases in Chinese LCQMC corpus which fine-tuned BERT makes mistakes on. The subscript of $IBH$ is consistent with the original label.

Lyu et al., 2021; Xu et al., 2022). The state-of-the-art methods achieve significant improvements, increasing PI performance up to the accuracy of 91.6% and 88.3% on the benchmark corpora QQP (Iyer et al., 2017) and LCQMC (Liu et al., 2018).

No matter whether in the English or Chinese scenario, the advanced PI models still suffer from two types of cases, as shown in Table 1. The first ones are classified as "**i**somorphic **b**ut **h**eterogeneous" (denoted as $IBH_0$), which is referred to the pair of sentences holding similar syntactic structure but with different semantics (label "0"). Cases of this type tend to be literally consistent (smaller edit distance), easily confusing models to identify such pairs as "Paraphrase". The second ones are classified as "**i**somerous **b**ut **h**omogeneous" (denoted as $IBH_1$). These cases are opposite of $IBH_0$, using different expressions to convey the same meaning (label "1"), which makes the model mispredict them as "Non-paraphrase" due to the huge literal differences. We conduct a pilot experiment on the QQP validation set. Among the incorrect predictions made by the fine-tuned BERT model, 35.9% of these cases correspond to $IBH_0$, while 14.7% correspond to $IBH_1$[1]. Errors in these two types occupy **50.7%** of all the errors made by the baseline model. It highlights the ineffectiveness of the

---

[1]We use a Levenshtein similarity with a threshold of 0.6 to identify literal similarity (sentences with a similarity greater than 0.6 are considered literally similar).

current PI model in handling these cases.

Advanced DA methods such as Adversarial Training have been proven to be effective in solving such cases (Hou et al., 2018; Kumar et al., 2020; Morris et al., 2020a; Li et al., 2022). However, advanced adversarial methods suffer from two limitations. The first one is inefficiency. The majority of these methods are based on greedy-based synonym replacement (Li et al., 2018; Ren et al., 2019; Jin et al., 2020; Li et al., 2020; Garg and Ramakrishnan, 2020). They first need to use external knowledge bases (e.g., WordNet) to obtain lists of synonyms. And after that, a greedy-based search strategy is used to choose suitable words, which brings serious time costs. The second one is poor quality. Previous work (Morris et al., 2020a) has proven most of them may result in generating unnatural and unreadable sentences, or even huge semantic shifts from the original ones.

With the continuous improvement of PLMs, generative models, such as BART (Lewis et al., 2020), are capable of generating sentences that are in line with human expression habits with acceptable time costs. Despite the fact that using seq2seq as a DA method often requires additional data or special data conditions, PI data has a naturally generative feature with source and target sentences, making seq2seq a great DA method.

We suggest that the existing PI models can be further improved in a simple and effective way by strengthening learning on misclassified instances. Inspired by the human learning process, we believe that, just as humans learn knowledge from books, the model aims to learn previously unknown knowledge from training data. For the knowledge that is difficult to understand, we need to improve our understanding and cognition through repeated and extended learning, and so are the models.

Therefore, in this study, we propose the Generative Boosting Training (GBT) method for single model, which utilizes seq2seq model as the DA method serving for our boosting learning algorithm. During the training process, GBT performs DA on the instances that the model did wrong in a period of training steps, and enforces the model to be re-trained on these instances. In this way, GBT can enable the model to correct its errors and enhance its ability to solve hard cases in a timely manner, just like human learning.

Experiments on the benchmarks of English QQP and Chinese LCQMC show that GBT improves var-ious PLM models. It is noteworthy that a baseline model (with only a PLM encoder and a linear classifier) can outperform the advanced models with the boosting of GBT. Our contributions can be summarized as follow:

- The proposed GBT brings stable and significant improvements over the current PLMs-based PI models.

- GBT utilizes the seq2seq model as the DA method, greatly improving the efficiency and quality of example generation.

- GBT is vest-pocket, which extends the training time by no more than 1 hour on an RTX3090 24GB GPU.

## 2 Related Work

**PI** has been widely studied due to its widespread application in downstream tasks. The existing approaches can be divided into representation-based and interaction-based approaches.

The representation-based approaches use a siamese architecture to encode two sentences into independent vectors of high-level features. On this basis, the semantic similarity is obtained by both feature vectors. He et al. (2015) propose a siamese CNN structure, and they verify the effectiveness of cosine similarity, Euclidean distance and element-wise difference. Wang et al. (2016) decompose two sentences to the sets of similar and different tokens to extract similar and distinguishable features by CNN. Lai et al. (2019) focus on the ambiguity caused by Chinese word segmentation and propose a lattice-based CNN model (LCNs). Lyu et al. (2021)'s LET-BERT perceives multi-granularity interaction based on word lattice graph, where graph networks and external knowledge *HowNet* (Dong and Dong, 2003) are used, achieving state-of-the-art performance in Chinese PI.

An interaction-based PI model is designed to perceive interaction features between sentences, and encode such information into the representations. Specifically, Wang et al. (2017) propose a BiMPM model under the matching-aggregation framework, which conducts one-to-many token-level matching operations to obtain interaction information. ESIM model proposed by Chen et al. (2017) performs interactive inference based on LSTM representations. ESIM sharpens the interaction information by means of element-wise dot product and subtraction. Inspired by ResNet (He et al., 2016a), Kim

et al. (2019) propose a densely-connected recurrent and co-attentive Network (DRCN), which combines residual operation with RNN and attention mechanism. Zhang et al. (2021) propose a Relation Learning Network ($R^2$-Net) based on BERT, which is characterized by interactive relationship perception of multi-granularity linguistic units. The recently proposed ISG-BERT (Xu et al., 2022) integrates syntactic alignments and semantic matching signals of two sentences into an association graph to gain a fine granularity matching process.

**DA** techniques such as Adversarial Training have received keen attention in solving semantic matching tasks. Ren et al. (2019) propose a Probability Weighted Word Saliency (PWWS), determining word substitution order by the word saliency and weighted by the classification probability, where WordNet (Miller, 1995) is used for synonym replacement. Similar to PWWS, Li et al. (2018) propose TextBugger, which finds word substitution order by Jacobian matrix and considers both character and token-level modification. The BERT-Attack proposed by Li et al. (2020) also applies a greedy word saliency strategy to find vulnerable words. It uses BERT-MLM for word generation instead of synonyms.

With the advancement of auto-regressive PLM, the seq2seq model is also used as a DA method in various scenarios (Hou et al., 2018; Kumar et al., 2020; Li et al., 2022). However, seq2seq is not generalized as a DA method such as synonym replacement, because it requires the data to have a generative condition, which is relatively strict. In this paper, we explore a generative augmentation method suitable for PI.

# 3 Approach

Let us first define the PI task in a formal way. Given a pair of sentences $X_1 = \{x_1^1, x_2^1...x_n^1\}$ and $X_2 = \{x_1^2, x_2^2...x_n^2\}$, a PI model aims to determine whether $X_1$ and $X_2$ are paraphrases of each other. During training, each data example can be noted as $(X_1, X_2, y)$, where y stands for a binary label indicating the relationship between $X_1$ and $X_2$: either "Paraphrase" (P) or "Non-paraphrase" (N). The goal of training is to optimize the PI model $f(X_1, X_2)$ by minimizing the prediction loss.

We suggest that the learning process of the model closely parallels that of humans. The goal is to learn previously unknown knowledge from one instance, reinforce impressions, and use the

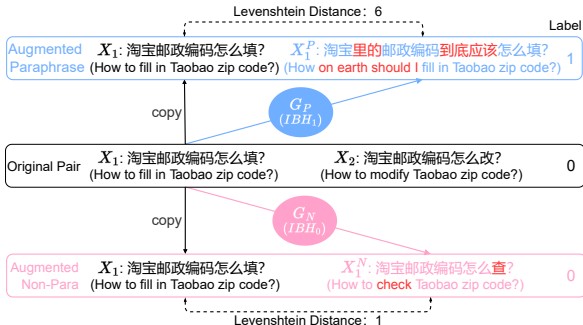

Figure 1: Utilizing $G_P$ and $G_N$ to perform DA to generate two augmented data (only on $X_1$).

learned knowledge to infer other instances. In addition, models need to timely correct mistakes during the learning process.

In light of this, we devise a simple but effective training method GBT to mimic the human learning process. First, we leverage the inherent properties of PI data to transfer it into seq2seq training data, enabling the training of two independent generators. During training, both generators need to perform DA on the cases where the model made mistakes. Subsequently, we re-train the model on the augmented data at a certain interval.

## 3.1 Baseline Model

As introduced by Devlin et al. (2019), we adopt the model framework of a PLM-based encoder with a two-layer classifier as our baseline model. The model is designed to predict the binary relationship of $X_1$ and $X_2$. The input sequence is organized as $X = \{[CLS], X_1, [SEP], X_2, [SEP]\}$ for BERT-based models and $X = \{, X_1, <\backslash s>, X_2, <\backslash s>\}$ for RoBERTa-based models, where $[CLS], [SEP],$ $$ and $<\backslash s>$ serve as special tokens. Further, we input X into the PLM encoder to produce its contextual representations of each token. The $[CLS]$ or $$ token is fed into linear classifier to estimate the probabilities $p(\hat{y}|X_1, X_2)$. The optimization objective is to minimize the cross-entropy loss:

$$L_{PI} = -(y * log(p(\hat{y}|X_1, X_2)) + (1-y) * log(1 - p(\hat{y}|X_1, X_2))). \quad (1)$$

## 3.2 Boosting Generator

For DA, we train two seq2seq models $G_P$ and $G_N$ as our generators. Given an input sentence, the $G_P$ aims to produce a paraphrased sentence that is literally different ($IBH_1$). While $G_N$ is to generate a non-paraphrased sentence that is literally similar to the input sentence ($IBH_0$).

**BART** ([Lewis et al., 2020](#)): BART is a classical seq2seq model using the standard Transformer structure including a bidirectional encoder and an autoregressive decoder. It is pre-trained by corrupting text with an arbitrary noising function, and learning to reconstruct the original text. We use BART as the backbone model for training both generators $G_P$ and $G_N$.

**Data Preparation**: We introduce the training data for two generators $G_P$ and $G_N$. First, we split the original PI training set into Paraphrase ($P$) and Non-Paraphrase ($N$) by the given binary label. We take $X_1$ as the source sentence and $X_2$ as the target sentence no matter on $P$ or $N$.

For data $P$, we aim to select paraphrased sentence pairs that $X_1$ and $X_2$ are literally different to train $G_p$. In this way, $G_p$ is able to learn how to generate paraphrased sentences with different syntactic structures from the source sentence. To be specific, we calculate the BLEU[2] score for each pair of $X_1$ and $X_2$ in data $P$. Then, we randomly select 50k examples from the top 50% of pairs with the lowest BLEU scores (a lower BLEU score indicates a greater dissimilarity). These selected examples are used as the training data for $G_P$.

While for data $N$, we select non-paraphrased sentence pairs that $X_1$ and $X_2$ are literally similar, which enables $G_N$ to produce target sentences with similar literal content but different semantics. For this, we also calculate the BLEU score of data $N$ and randomly select 50k examples from the top 50% of pairs with the highest BLEU score as training data for $G_N$.

**Training Process**: We train the generators $G_P$ and $G_N$ by fine-tuning the backbone model BART. Specifically, given training data ($X_1$, $X_2$), the source sentence $X_1$ is fed into the encoder, and the decoder is required to generate tokens in target sentence $X_2$ auto-regressively. The $G_\theta$ is trained to maximize the log-likelihood:

$$L_{Gen}(\theta) = \sum_{t=1}^{T} log(p_\theta(x_t^2|x_{1:t-1}^2, X_1)) \quad (2)$$

where $x_t^2$ represents current generating word in $X_2$, and $x_{1:t-1}^2$ represents previous word in the ground-truth rather than word $\hat{x}_{1:t-1}^2$ generated by the model. This training technique is known as teacher forcing.

---

[2]Our choice of BLEU, instead of other evaluation metrics, is motivated by our specific aim to measure literal similarity. In this context, BLEU adequately satisfies our purpose and provides computational efficiency.

The generator is evaluated by the BLEU metric. The final BLEU value for fine-tuned Chinese generator is 51.5 ($G_P$) and 48.8 ($G_N$), while that for the English generator is 29.6 ($G_P$) and 27.7 ($G_N$). After training, the parameters of $G_P$ and $G_N$ are fixed for subsequent use, noted as Boosting Generator.

### 3.3 Boosting Training

Classical boosting learning methods perform ensemble learning. During training time, the integrated model serially adds new models to itself to strengthen the cases where the previous integrated model did wrong. In this way, the integrated model is able to enhance the ability to handle hard cases.

Different from the classical boosting method, GBT repeatedly enhances the ability of a single model to solve hard cases without the massive cost of multiple models. In a round of learning, GBT records wrong instances during fixed $t$ training steps (denoted as boosting interval $t$). The boosting interval $t$ controls how often the model will reinforce learning on the wrong instances. In order to be corrected in time, the interval t should not be too large. Because the model will have a large deviation from the previous model that did wrong at that time due to parameter updating, the effect of re-learning wrong instances will also decrease. After every $t$ learning steps, the model will enter boosting training mode from the normal training.

In boosting training mode, wrong cases will be augmented by $G_P$ and $G_N$ as shown in Figure 1. Note that different from Figure 1, we perform DA not only on $X_1$ but also $X_2$. Given a wrong instance ($X_1$, $X_2$, y), we will generate four extra cases according to it. First, $G_P$ produces paraphrase sentences of $X_1$ and $X_2$ respectively, and we note them as $X_1^P$ and $X_2^P$. $G_N$ is also used to generate non-paraphrase sentences of $X_1$ and $X_2$, denoting as $X_1^N$ and $X_2^N$. In this way, we are able to produce two $IBH_1$ cases ($X_1$, $X_1^P$, 1) and ($X_2$, $X_2^P$, 1), also two $IBH_0$ cases ($X_1$, $X_1^N$, 0) and ($X_2$, $X_2^N$, 0) according to one wrong instance.

The GBT learning process is shown in Algorithm 1. Note that we randomly select p% cases for boosting training (L9 in Algorithm 1). This is because the examples we generate are relatively difficult, and the proportion of difficult examples should not be too high. Besides, we did not start the first boosting training until the warm-up steps ends (L8 in Algorithm 1). We will discuss the boosting interval $t$, boosting ratio $p$, and the starting timing

**Algorithm 1:** Boosting Training

**Input:** original example $X = [X_1; X_2]$,
label $y$, boosting generator $G_P(\cdot)$,
$G_N(\cdot)$, boosting interval $t$, boosting
ratio $p$, target model $f(\cdot)$, warm-up
steps $ws$, training function **train**($\cdot$)

**Output:** None

1   Define train_steps $ts$, $W$ of wrong cases set;
2   Begin training;
3     **For** $X$ in TrainData **Do**:
4       $ts$ += 1
5       **train**($f, X$)
6       **If** $f(X)! = y$ **Do**:
7         W.add(X)
8       **If** $ts\%t == 0$ and $ts >= ws$ **Do**:
9         W –> randomly save p% cases
10       **Boosting_Train**($f, W$)
11       W.clear()
12   **Boosting_Train**($f, W$):
13     **For** $X$ in $W$ **Do**:
14       $G_P(X_1) \rightarrow X_1^P$     $G_P(X_2) \rightarrow X_2^P$
15       $G_N(X_1) \rightarrow X_1^N$     $G_N(X_2) \rightarrow X_2^N$
16       $W$.add($(X_1, X_1^P, 1), (X_2, X_2^P, 1)$)
17       $W$.add($(X_1, X_1^N, 0), (X_2, X_2^N, 0)$)
18     **train**($f, W$)

| Dataset | Size | Pos:Neg | Domain |
|---------|------|---------|--------|
| LCQMC | 260,068 | 1:0.7 | Open-Domain |
| QQP | 404,276 | 1:2 | Open-Domain |

Table 2: Statistics for LCQMC and QQP

### 4.2 Hyperparameter Settings

Our experiments are conducted on HuggingFace's
Transformers Library. We adopt an Adam opti-
mizer with epsilon of 1e-8 and warm-up steps ratio
of 10% for all tasks. For the generative model, we
first fine-tune the BART$large$ models (the Chinese
version is provided by fnlp[3]) on both benchmarks
(Section 3.2). The batch size and learning rate are
set to 16 and 2e-5. We set the epoch to 3 and the
max sequence length is set to 100.

For Chinese PI tasks, we set the learning rate to
2e-5 and batch size to 32 for LCQMC. While for
the English task, the learning rate and batch size
are set to 2.5e-5 and 64. The fine-tuning epoch is
set to 5 for LCQMC. While for QQP, we fine-tune
the models for 50k steps, and checkpoints are eval-
uated every 2k steps. We set the boosting interval
$t = 500$ steps and boosting ratio $p = 25(\%)$. All
experiments are conducted on a single RTX 3090.

### 4.3 Main Results

Table 3 and Table 4 show the main results on
LCQMC and QQP. The reported performance of
our models are average scores obtained in five runs
with random seeds.

**LCQMC**: The Chinese PI models are listed in
Table 3. Our proposed GBT yields significant im-
provements (p-value < 0.05 in statistical signifi-
cance test) over baselines, which demonstrates that
GBT generalizes well when cooperating with dif-
ferent PLMs. The most substantial boosting is up
to 3.5% ($Acc.$) and 2.5% ($F_1$), achieving state-of-
the-art performance.

In addition, our models outperform all state-of-
the-art models, including the recently-proposed
LET-BERT which addresses issues of complex Chi-
nese word segmentation and ambiguity to some
extent. It uses $HowNet$ as the external knowledge
base to provide explanations. While for GBT, it is
noteworthy that we use no extra knowledge. GBT
takes advantage of the existing data to train the
Boosting Generator, and only costs some additional
time for DA and boosting training during the train-
ing process. In this way, a baseline model without

in Section 4.5.

The model is required to be re-trained on the
wrong instances and their augmented ones to en-
hance performances. After that, the model will
return to normal training steps to continue a new
round of learning.

## 4 Experimentation

### 4.1 Corpora and Evaluation Metrics

We evaluate our PI models on two benchmark cor-
pora, including Chinese LCQMC (Liu et al., 2018),
as well as English QQP (Iyer et al., 2017). Both
QQP and LCQMC are large-scale corpora for open-
domain, where sentence pairs are collected from
QA websites without rigorous selection. Each in-
stance in both corpora is specified as a pair of sen-
tences. The binary labels of "Pos" (i.e., paraphrase)
and "Neg" (i.e., non-paraphrase) are provided. Ta-
ble 2 shows the statistical information of the cor-
pora. The canonical splitting method of training,
validation and test sets for the corpora is plainly
stated, and we strictly adhere to it. We evaluate
all the models in the experiments using accuracy
(ACC.) and F1-score.

---

[3] https://nlp.fudan.edu.cn

| Models | LCQMC | |
|---|---|---|
| | $ACC.$ | F1 |
| Text-CNN (He et al., 2016b) | 72.8 | 75.7 |
| BiLSTM (Mueller and Thyagarajan, 2016) | 76.1 | 78.9 |
| Lattice-CNN (Lai et al., 2019) | 82.1 | 82.4 |
| BiMPM (Wang et al., 2017) | 83.3 | 84.9 |
| ESIM (Chen et al., 2017) | 82.5 | 84.4 |
| GMN-BERT (Chen et al., 2020) | 87.3 | 88.0 |
| LET-BERT (Lyu et al., 2021) | 88.3 | 88.8 |
| BERT (Devlin et al., 2019)$^\diamond$ | 85.7 | 86.8 |
| + GBT♣ | 89.2 | 89.3 |
| BERT-wwm (Cui et al., 2021)$^\diamond$ | 86.8 | 87.7 |
| + GBT ♣ | 89.4 | 89.5 |
| MacBERT (Cui et al., 2021)$^\diamond$ | 87.0 | 88.0 |
| + GBT♣ | 89.6 | 89.7 |
| MacBERT$_{large}$ (Cui et al., 2021)$^\diamond$ | 87.6 | 88.3 |
| + GBT♣ | **89.9** | **89.9** |

Table 3: Comparison to the performance (%) of baselines and previous work on Chinese LCQMC. The mark "$\diamond$" denotes the PLM-based baselines, while "♣" denotes GBT boosted models that obtain significant improvements (p-value $< 0.05$ in statistical significance test) over baselines.

| Models | QQP | |
|---|---|---|
| | $ACC.$ | F1 |
| CENN (Zhang et al., 2017) | 80.7 | \ |
| L.D.C (Wang et al., 2016) | 85.6 | \ |
| BiMPM (Wang et al., 2017) | 88.2 | \ |
| DIIN (Gong et al., 2017) | 89.1 | \ |
| DRCN (Kim et al., 2019) | 90.2 | \ |
| $R^2$-Net (Zhang et al., 2021) | 91.6 | \ |
| ISG (RoBERTa-$large$) (Xu et al., 2022) | 91.4 | \ |
| BERT (Devlin et al., 2019)$^\diamond$ | 89.4 | 87.5 |
| + GBT♣ | 91.5 | 88.7 |
| BERT$_{large}$ (Devlin et al., 2019)$^\diamond$ | 89.6 | 87.8 |
| + GBT♣ | 91.7 | 88.9 |
| RoBERTa (Liu et al., 2019)$^\diamond$ | 89.9 | 88.4 |
| + GBT♣ | 91.8 | 89.0 |
| RoBERTa$_{large}$ (Liu et al., 2019)$^\diamond$ | 90.0 | 89.1 |
| + GBT♣ | **92.3** | **89.6** |

Table 4: Comparison to the performance (%) of baselines and previous work on English QQP. The mark "$\diamond$" denotes the PLM-based baselines, while "♣" denotes GBT boosted models that obtain significant improvements (p-value $< 0.05$ in statistical significance test) over baselines.

any enhancing component network can outperform the existing strongest model.

**Quora**: Table 4 shows the performance of English PI on QQP. The previous work didn't report F1-scores on QQP. We evaluate our models with the F1 metric and report the performance to support future comparative studies. Similarly, GBT boosts baseline models stably on different configurations and PLMs, achieving state-of-the-art performance.

The previous approach $R^2$-Net is strongest among the advanced models. Its structure contains both CNN and PLM-based encoder. Besides, it is learned by utilizing the label information of instances, and additionally trained by congeniality recognition among multiple instances. Our approach achieves comparable performance to $R^2$-Net when BERT$base$ is used as the backbone, but with no sophisticated structure and learning process. The recently proposed ISG model is another powerful model, using a graph model to integrate syntactic alignments and semantic matching signals. Our model outperforms ISG when RoBERTa$large$ is used with no extra parameters and structure. In the English scenario, GBT also brings strong improvements over the baseline models to achieve state-of-the-art performance.

We can observe that improvements in Chinese PI are more than in English. This is partly due to the effect of generator models under different language scenarios. The generation length in Chinese is shorter and the difficulty is lower, which makes the generation effect better. Therefore, the boosting effect for Chinese PI is better than that of English.

### 4.4 Ablation Study

We conduct ablation experiments over the baseline model using BERT as described in Section 3.1. Three expanded models are considered in this experiment. First, "+Only Boosting Generator" uses Boosting Generator to generate augmented cases on all training data no matter whether the model did it right or wrong, which makes the training data four times larger than before. Second, "+ Only Boosting Train" applies the Boosting Training algorithm in Algorithm 1 but only re-train on the original wrong cases with no Boosting Generator for DA. Last, the GBT combines both aforementioned approaches to enforce the GBT approach.

Table 5 shows the performance obtained on test sets. It is found that the performance has sharply dropped with Boosting Generator performing on the full amount of data. This is due to the unbalanced proportion of difficult instances and redundant samples. While only applying Boosting Training simply on the original wrong cases brings substantial improvements. This demonstrates that despite not utilizing Boosting Generator for DA, our boosting training algorithm is an effective means of improving performance.

Last, we prove that GBT combines Boosting

| Models | LCQMC | | QQP | |
|---|---|---|---|---|
| | ACC. | F1 | ACC. | F1 |
| BERT | 85.7 | 86.8 | 89.4 | 87.5 |
| + Only Boosting Generator | 79.4 | 81.0 | 81.2 | 83.1 |
| + Only Boosting Training | 88.2 | 88.6 | 90.9 | 88.1 |
| + GBT | **89.2** | **89.3** | **91.5** | **88.7** |

Table 5: Ablation study on both benchmarks.

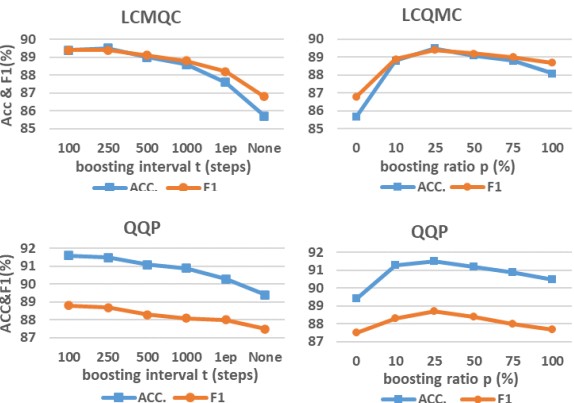

Figure 2: Effectiveness of boosting interval $t$ and boosting ratio $p$ on both benchmarks.

Training with Boosting Generator to achieve further improvements. Boosting Training algorithm finds suitable instances and timing for Boosting Generator to make DA, while the latter brings stronger improvements.

## 4.5 Boosting Timing and Ratio

As mentioned in Algorithm 1, we use boosting interval $t$ to control the boosting timing, and boosting ratio $p\%$ to control how many instances should be augmented for difficulty balance. Figure 2 shows how the interval $t$ and ratio $p$ affect model learning.

We conduct experiments when the boosting interval $t$ is in $\{100, 250, 500, 1000, 1ep\}$, where $1ep$ means we perform the boosting learning at the end of each epoch. As expected, the smaller the interval $t$ brings better performance, because the model can correct mistakes and learn expanded knowledge (augmented examples) in a timely manner. However, frequent switching of training modes will lead to a certain amount of extra cost. Therefore, we believe $t = 250$ is a good choice.

Moreover, we also test the boosting ratio $p$ on both benchmarks where $p$ takes values from $\{10, 25, 50, 75, 100\}$ (%). We set boosting ratio $p = 25\%$ as a reasonable value. It can be observed that if $p$ is too small, the enhanced effect will not be obvious. While if $p$ is too large, it will lead to an imbalance in the proportion of difficult examples, eventually affecting the performances.

Just like the human learning process, it is better to learn difficult knowledge after mastering a certain foundation instead of learning difficulties at the beginning. Therefore, we test three timings on when we should start **first** boosting training, respectively at 1) the beginning of the whole training, 2) after the first epoch (start at the second epoch) 3) after the end of the warm-up steps. As expected, start timing at 1) performs worst and 3) works best. That is why we do not start boosting training until the warm-up steps end (L8 in Algorithm 1).

## 4.6 Comparison with advanced DA methods

Compared with advanced DA methods, GBT has the advantage of generating high-quality instances with low time cost because we use the seq2seq model (Boosting Generator). We replace the Boosting Generator with advanced DA methods to analyze their efficiency and performances as shown in Table 6. All adversarial methods we used as baselines are open-source and readily applicable to the PI task. We implement these methods using TextAttack[4] (Morris et al., 2020b). We conduct experiments on the QQP dataset, because under the Chinese scenario, there are few DA-related researches that we can compare with. We compare our models to a series of strong arts, including:

- **PWWS** (Ren et al., 2019): Based on the synonym replacement strategy, PWWS proposes a probability-weighted word salience approach to determine word replacement order. The synonyms are provided by WordNet.

- **TextBugger** (Li et al., 2018): TextBugger finds important words by Jacobian matrix and greedy search. In order to ensure that the generated examples are literally and semantically consistent with the original ones, character and word-level modifications are considered.

- **BERT-Attack (BERTA)** (Li et al., 2020): BERTA identifies vulnerable words by loss-based scores, which is to mask tokens one by one to see the loss changes. The BERT MLM is used to generate top-$K$ candidates for each keyword, and another greedy algorithm is applied to confirm the final option. Note that

---

[4] https://github.com/QData/TextAttack

| Models | QQP | | Boosting instances | Time cost | Generation method |
|---|---|---|---|---|---|
| | $ACC.$ | F1 | (total) | (sec. per instance) | |
| BERT (Devlin et al., 2019) | 89.4 | 87.5 | 0 | 0 | None |
| GBT (PWWS) (Ren et al., 2019) | 89.9 | 87.9 | 20984 | 0.42 (699.7%) | Greedy + Synonym |
| GBT (TextBugger) (Li et al., 2018) | 90.5 | 88.4 | 20700 | 0.21 (350.0%) | Greedy + Gradient |
| GBT (BERTA) (Li et al., 2020) | 90.8 | 88.4 | 21132 | 0.61 (1016.7%) | Greedy + MLM |
| **GBT$_{ours}$ (Boosting Generator)** | **91.5** | **88.7** | **15144** | **0.06** (100.0%) | **Seq2seq (BART**$large$**)** |

Table 6: Time efficiency and generated effectiveness comparison between our Boosting Generator and advanced DA methods on English QQP.

default $K$ is set as 48, but the time cost is way too much and we regulate $K$ as 8.

As mentioned, most previous DA methods are based on greedy search for word replacements, and there are also a few methods using MLM for word generation. However, most methods have been criticized for their efficiency or generation quality shortcomings (Morris et al., 2020a). With the progress of the generative model, we believe that the generative model is far better than rule-based methods under certain conditions, and it can perform well in sentence diversity and fluency.

As shown in Table 6, GBT has a much higher generation efficiency in *Time cost*. Our GBT with seq2seq generator is several times more efficient than advanced methods, proving that the generative model is able to achieve the expected effect. The generation quality can be evaluated through the comparison in performance. The model training on the instances generated by our Boosting Generator brings the strongest improvements. Meanwhile, GBT with our Boosting Generator produces fewer boosting instances, proving that our model has relatively fewer wrong instances during training.

In addition, it can also be found that despite replacing different DA methods, the model still gains substantial improvements, demonstrating the generalization performance of our Boosting Training algorithm. Moreover, when the DA method is replaced, our GBT is not limited to the PI task but can be generalized to any task.

## 4.7 Case study

GBT provides a time-efficient DA method as described in Table 6. Compared with other DA methods, the performance improvement of GBT proves the excellent generation quality to a certain extent. We show an English case in QQP generated with different DA methods in Table 7.

As shown, four examples are produced by using $G_P$ and $G_N$ to generate from $X_1$ and $X_2$ respec-

| | Sentences | Label |
|---|---|---|
| Original | $X_1$: How do I remove a wart? | 1 |
| | $X_2$: How do I remove wart on my face? | |
| PWWS | $X_1$: How do I slay a wart? | 1 |
| | $X_2$: How do I remove wart on my face? | |
| TextBugger | $X_1$: How do I remove a wart? | 1 |
| | $X_2$: How do I eliminate wart on my face? | |
| BERTA | $X_1$: How do I replace a wart? | 1 |
| | $X_2$: How do I erase wart on my face? | |
| $G_P(X_1)$ | $X_1$: How do I remove a wart? | 1 |
| $(IBH_1)$ | $X_1^P$: What is the best way to remove wart? | |
| $G_N(X_1)$ | $X_1$: How do I remove a wart? | 0 |
| $(IBH_0)$ | $X_1^N$: Why do you remove a wart? | |
| $G_P(X_2)$ | $X_2$: How do I remove wart on my face? | 1 |
| $(IBH_1)$ | $X_2^P$: What is the best way to remove wart on face? | |
| $G_N(X_2)$ | $X_2$: How do I remove wart on my face? | 0 |
| $(IBH_0)$ | $X_2^N$: Why do you remove a wart on your face? | |

Table 7: Examples of data augmentation with GBT and other DA methods.

tively (as described in Section 3.3). Our generated sentence is marked in blue. While the modifications of other DA methods are marked in red.

PWWS and TextBugger replace the word *"remove"* with *"slay"* and *"eliminate"* respectively. Although the generated sentence is semantically similar to the original sentence, it is unnatural and does not conform to normal pragmatic habits. BERTA replaces *"remove"* in both sentences with *"replace"* and *"erase"*, resulting in semantic shifts from the original pair. Moreover, all three methods generate "Paraphrase" sentences that are literally similar to the original ones, which makes the model easy to recognize.

It can be observed that our generative approach can not only generate "Non-Paraphrase" examples but also produce more natural and diverse results. When generating "Paraphrase", it recognizes the semantics between *"How"* and *"What is the best way"* are similar, changing the syntactic structure while preserving the same meaning ($IBH_1$). While for "Non-paraphrase", it switches *How-type* questions to *Why-type* questions, as to preserve similar literal structure while changing the semantics ($IBH_0$). In this way, our GBT is able to bring better effectiveness during the training process.

## 5 Conclusion

In this study, we propose a Generative Boosting Training (GBT) approach for PI. GBT combines the generative DA method with our boosting learning algorithm, which enables the model to correct mistakes and perform expanded learning in a timely manner during the learning process. GBT offers several advantages: high-quality and efficient DA for PI, simple and effective boosting learning algorithm, and state-of-the-art matching performance. Experimental results on both English and Chinese benchmarks verified the effectiveness and efficiency of GBT. In the future, we will explore more generalized generative DA methods to broaden the application of GBT to various tasks.

## Limitations

GBT leverages the seq2seq model as a high-quality and efficient DA method. However, the seq2seq method requires data with the generative property (such as data for PI), which limits the generalization of our DA method to other tasks. In contrast, our boosting learning algorithm is a method that can be generalized to other tasks. Given the remarkable ability of Large Language Models (LLMs) to adhere to instructions, we can use LLMs to generate data tailored to the specific task by providing well-designed prompts. In this way, our GBT can achieve generalization to other tasks.

Furthermore, GBT may not handle noisy examples with incorrect labels well, as it tends to repeatedly emphasize the influence of such examples during the training stage. This issue of mislabeling can be mitigated through data-cleaning approaches.

## Acknowledgements

We thank all anonymous reviewers for their insightful comments. This work is supported by National Key R&D Program of China (2020YFB1313601) and National Science Foundation of China (62376182, 62076174).

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
