# OpenReview forum: "GBT: Generative Boosting Training Approach for Paraphrase Identification"
_EMNLP/2023/Conference — EMNLP 2023 Findings_

### Official Review · Reviewer_xb7T · 2023-08-04

**Typos Grammar Style And Presentation Improvements:** 1)	Table 4 uses the same symbols as t…
**Soundness:** 2

**Excitement:**

2: Mediocre: This paper makes marginal contributions (vs non-contemporaneous work), so I would rather not see it in the conference.

**Missing References:**

There are a few missing citations throughout the draft. Some of them are as follows:
1) text in line 92-98
2) In line 299-300


**Paper Topic And Main Contributions:**

The work proposes a technique named Generative Boosting Training (GBT) where additional synthetic data is generated for difficult cases (where model makes mistake in prediction) on the task of paraphrase identification (PI). Authors propose to augment paraphrase sentences that are different in text but have similar semantic meaning (referred to as IBH1) and sentences that have similar text but different semantic meaning (referred to as IBH0). The work shows improvement from on 2 PI datasets, PPQ and LCQMC.

**Questions For The Authors:**

1)	How much was the benefit from just doing DA with IBH1 instances?

**Reasons To Accept:**

1)	The GBT technique shows improvement (as per the result tables 3 and 4) and could be applied generally to other tasks as well. The negative and positive paraphrase augmentation seems like a good mix of balanced data augmentation (though we don’t know how much just adding IBH1 and just adding IBH0 independently contribute to improvement).
2)	The propose technique seems like a good choice for the PI task and some of the analyses (Figure 2 for boosting interval and ratio, table 5 for why to use GBT for hard examples) highlight interesting takeaways about GBT.

**Reasons To Reject:**

1)	There are potentially numerous baselines as data augmentation for hard examples has several work. Given the closeness with this proposed work of using paraphrases (both negative and positive), some of baselines are necessary for comparison with GBT, especially counterfactual data-augmentation techniques as GBT uses (negative paraphrases in DA i.e., IBH0)
Feng et.al., A Survey of Data Augmentation Approaches for NLP
Li  et.al., Data Augmentation Approaches in Natural Language Processing: A Survey
2)	Some other, even more simpler baselines could be lower learning rate and training for more number of epochs. This would even strengthen the claims of GBT if improvements are significant.
3)	The ablation study in table 5 seemed more like good baselines which is good to have as it shows GBT is more effective when applied to only hard examples. A better ablation study could be giving just positive paraphrases (IBH1) and just negative paraphrases (IBH0)
4)	Some of the important technical details are unclear. For example, which datasets and how was the paraphraser trained to generate candidate sentences for selecting IBH0 and IBH1. In line 268-277, more details would be needed as to how and where the 50K examples were selected from.
5)	The text in line 293-295 makes the above point a little bit more unclear. It would be difficult for readers to understand and evaluate – “we manually observed the generated examples and find the results acceptable.”
6)	A very minute point – it may be interesting to compare with openLLM methods like LLaMa (after some instruction tuning for PI  task).

**Reproducibility:**

3: Could reproduce the results with some difficulty. The settings of parameters are underspecified or subjectively determined; the training/evaluation data are not widely available.

**Reviewer Confidence:**

4: Quite sure. I tried to check the important points carefully. It's unlikely, though conceivable, that I missed something that should affect my ratings.

---

> ### Author Rebuttal · Authors · 2023-08-28
>
> We extend our sincere appreciation for your dedicated effort in reviewing our papers and offering valuable suggestions to enhance their quality. Your feedback holds significant value for us, and we genuinely consider each suggestion you provide. We take your feedback seriously, and our response is as follows:
>
>
>
> **----------Response to Comments----------**
>
>
> **Comment #1:** There are potentially numerous baselines as data augmentation for hard examples has several work. Given the closeness with this proposed work of using paraphrases (both negative and positive), some of baselines are necessary for comparison with GBT, especially counterfactual data-augmentation techniques as GBT uses (negative paraphrases in DA i.e., IBH0) Feng et.al., A Survey of Data Augmentation Approaches for NLP Li et.al., Data Augmentation Approaches in Natural Language Processing: A Survey
>
>
> **Response to #1:** Thank you for your valuable feedback. The data augmentation methods you mentioned are indeed valuable references. For comparison, the methods we used as baselines are open-source and readily applicable to the PI task. We are also open to incorporating other techniques, such as using LLMs for data augmentation. We will consider including them in our future experiments.
>
>
> **Comment #2:** Some other, even more simpler baselines could be lower learning rate and training for more number of epochs. This would even strengthen the claims of GBT if improvements are significant.
>
>
> **Response to #2:** Your suggestion presents a valuable comparative baseline. However, in our study, we have opted for the same learning rate and number of training epochs as the previous work and baseline models, so as to ensure a fair comparison. We appreciate your feedback and will include relevant experiments in our future work.
>
>
> **Comment #3:** The ablation study in table 5 seemed more like good baselines which is good to have as it shows GBT is more effective when applied to only hard examples. A better ablation study could be giving just positive paraphrases (IBH1) and just negative paraphrases (IBH0)
>
>
> **Response to #3:** Your suggestion is insightful. In our ablation study, we treated the combination of the two major components of our method (boosting generator and boosting training method) as a unified model first for comprehensive performance verification, and then investigate their respective contributions independently. If space permits, we will discuss the impact of different class data proportions on the experimental results in a separate section, as you have suggested.
>
>
> **Comment #4:** Some of the important technical details are unclear. For example, which datasets and how was the paraphraser trained to generate candidate sentences for selecting IBH0 and IBH1. In line 268-277, more details would be needed as to how and where the 50K examples were selected from.
>
>
> **Response to #4:** In line with your valuable feedback, we appreciate the opportunity to clarify some of the technical details. In our work, as mentioned in line 257, we utilized the training data from paraphrase identification, which corresponds to the training set of datasets LCQMC or QQP used in this work, for training the generator.
>
> Regarding the selection of the 50k examples, we follow a specific criterion, as explained in lines 265-270 and line 274-277. For paraphrased sentence pairs, we randomly sampled 50k pairs from the top 50% with the highest BLEU scores. Conversely, for non-paraphrased sentence pairs, we randomly selected 50k pairs from the top 50% with the lowest BLEU scores. This data selection process yields two distinct sets of examples, aligning with the two characteristics we previously mentioned. These two sets of data were then used to train two separate generators, with the training process elaborated upon in lines 278-288.
>
>
> **Comment #5:** The text in line 293-295 makes the above point a little bit more unclear. It would be difficult for readers to understand and evaluate – “we manually observed the generated examples and find the results acceptable.”
>
>
> **Response to #5:** We conducted a manual evaluation of the generator's performance, which, due to page limit, was not included in the paper. Specifically, we randomly sampled 500 data instances generated by the generator and subjected them to human assessment. For positive samples, our evaluation criterion was that two sentences were lexically similar (differing only in a few words) but semantically dissimilar. Conversely, for negative samples, we evaluated instances where the sentences were lexically dissimilar (with only a few shared words) but semantically similar. The proportions of qualified positive and negative examples are shown in the following table. They demonstrate that the generator's performance aligns well with our requirements. Notably, the accuracy of the negative sample generator is relatively lower, primarily because this task poses a higher level of difficulty, requiring the model to generate sentences that are lexically similar but semantically dissimilar. Exemplar can be found in Table 7 in the submission.
>
> |                    | Positive Pairs | Negative Pairs | All Pairs |
> | ------------------ | -------------- | -------------- | --------- |
> | Qualified          | 228            | 212            | 440         |
> | Total Number       | 250            | 250            | 500       |
> | Qualification Rate | 91.2%          | 84.8%          | 88%       |
>
>
> **Comment #6:** A very minute point – it may be interesting to compare with openLLM methods like LLaMa (after some instruction tuning for PI task).
>
>
> **Response to #6:** This is a valuable suggestion. The reason we didn’t conduct this experiment was to make a direct comparison with the existing studies. The openLLM methods, such as LLaMa, indeed present an interesting avenue for additional investigation. We will carry out the experiments for comparison in the extended version of our paper.
>
>
>
> **----------Response to Questions----------**
>
>
> **Question #1:** How much was the benefit from just doing DA with IBH1 instances?
>
>
> **Response to #1:** When performing DA exclusively on IBH1 instances, our proposed GBT method achieved an accuracy of 89.0% and an F1 score of 89.1% on LCQMC dataset. This performance slightly decreased compared to augmenting both IBH1 and IBH0 data, where the accuracy was 89.2%, and the F1 score was 89.3%. This underscores the necessity of augmenting both classes of data, as it enables the model to learn from a broader set of challenging samples. Furthermore, augmenting only IBH1 instances outperformed the baseline with no data augmentation, which has an accuracy of 88.2% and an F1 score of 88.6%. This highlights the significance of extending learning to augmented data, thereby enhancing the model's performance on unseen data.
>
>
>
> **----------Response to Typos Grammar Style And Presentation Improvements----------**
>
>
> We sincerely appreciate your valuable feedback on our writing, and we are committed to incorporating your suggestions into the revised version of the paper to enhance its overall quality.
>
>
> Thank you again for your insightful feedback, which undoubtedly contributes to enhancing the quality and enriching the final version of the paper.

---

### Official Review · Reviewer_Zs1Z · 2023-08-05

**Typos Grammar Style And Presentation Improvements:** 1. Figure 1
**Soundness:** 4

**Excitement:**

4: Strong: This paper deepens the understanding of some phenomenon or lowers the barriers to an existing research direction.

**Paper Topic And Main Contributions:**

This paper introduces a generative boosting training (GBT) method to address the paraphrase identification (PI) task. In detail, the authors use seq2seq model e.g. BART, to train two data augmentators for paraphrase ad non-paraphrase respectively. Then, these two types of data are augmented when having wrong case during normal training and thus are utilized to retrain the model.

**Questions For The Authors:**

A. Why did you use LCQMC and QQP as metrics?
B. Considering BLEU is not a perfect metric, have you thought of using other metrics to quantify the differences?

**Reasons To Accept:**

1. Good motivation to solve the paraphrase identification, with current limitations, detailed examples and vivid diagrams
2. Solid experiment on both BERT and GPT, reaching good performance or even state-of-the-art compared to the baseline models.
3. Explore the detailed of such data augmented boosting training, including but no limited to, boosting time, boosting ratio, different language models, etc.
4. GBT is efficient since it can be completed 1-hour on GPU.

**Reasons To Reject:**

1. The advanced data augmentation methods in comparison are from 2018-2020. It is unclear that whether the paper is comparing with the stat-of-the-art counterparts.
2. It is not sure that whether part of the performance improvement can be attributed to the learned language ability from the seq2seq model via generated data (like the language model distillation), but not the GBT.

**Reproducibility:**

4: Could mostly reproduce the results, but there may be some variation because of sample variance or minor variations in their interpretation of the protocol or method.

**Reviewer Confidence:**

3: Pretty sure, but there's a chance I missed something. Although I have a good feel for this area in general, I did not carefully check the paper's details, e.g., the math, experimental design, or novelty.

---

> ### Author Rebuttal · Authors · 2023-08-28
>
> We extend our sincere gratitude for dedicating your valuable time to reviewing our paper and providing us with invaluable insights and suggestions. Your feedback is greatly appreciated and has undoubtedly contributed to the refinement of our work. We have meticulously considered each of your suggestions and comments. We answer the concerns as below:
>
>
>
> **----------Response to Comments----------**
>
>
> **Comment #1:** The advanced data augmentation methods in comparison are from 2018-2020. It is unclear that whether the paper is comparing with the stat-of-the-art counterparts.
>
>
> **Response to #1:** We totally agree with your opinion. Numerous data augmentation methods exist across various tasks. However, our objective is to generate two distinct categories of challenging samples, including the ones that exhibit literal similarity but possess different semantics (IBH0), and the cases which lack literal similarity while sharing same semantics (IBH1). To the best of our knowledge, there is a limited availability of open-source data augmentation methods of this nature. The data augmentation methods that we have compared are widely employed and adept at generating the required data.
>
>
> **Comment #2:** It is not sure that whether part of the performance improvement can be attributed to the learned language ability from the seq2seq model via generated data (like the language model distillation), but not the GBT.
>
>
> **Response to #2:** Regarding this concern, we conducted an ablation experiment in Section 4.4 and the results can be found in Table 5, where we compare the performance of the baseline model with the ' + Only Boosting Generator' variant. When we train the model solely on data generated by the seq2seq model without using boosting training, we observe a significant drop in model performance. This demonstrates that relying solely on a seq2seq model is insufficient.
>
>
>
> **----------Response to Questions----------**
>
>
> **Question #1:** Why did you use LCQMC and QQP as metrics?
>
>
> **Response to #1:** We chose to use these two datasets primarily because they are well-known benchmarks in paraphrase identification, and a large amount of prior studies has been conducted on them [1][2]. Our experiments on these two benchmarks provide better comparability with previous work. The utilization of ACC and F1 as evaluation metrics also follows established practices in the previous work. We provided an introduction to both of these datasets and the metrics used in Section 4.1.
>
>
> **Question #2:** Considering BLEU is not a perfect metric, have you thought of using other metrics to quantify the differences?
>
>
> **Response to #2:** Indeed, there are several metrics, such as BERTScore [3], that are considered better than BLEU for measuring the similarity between two sentences. However, our specific objective revolves around the identification of **literal similarity** or dissimilarity within sentences. In this context, the BLEU metric effectively fulfills our intended purpose, offering the added advantage of computational simplicity. In our future work, we plan to investigate the impact of different evaluation metrics on our approach.
>
>
> **----------Response to Typos Grammar Style And Presentation Improvements----------**
>
>
> We sincerely appreciate your valuable feedback on our writing, and we are committed to incorporating your suggestions into the revised version of the paper to enhance its overall quality.
>
>
>
> Thank you again for your insightful feedback, which undoubtedly contributes to enhancing the quality and enriching the final version of the paper.
>
>
>
> **----------Reference----------**
>
>
> [1] Boer Lyu, Lu Chen, Su Zhu, and Kai Yu. 2021. Let: Linguistic knowledge enhanced graph transformer for chinese short text matching. In Proceedings of the AAAI, volume 35, pages 13498–13506.
>
> [2] Qiwei Peng, David Weir, Julie Weeds, Yekun Chai. 2022. Predicate-argument based bi-encoder for paraphrase identification. In Proceedings of the 60th Annual Meeting of the Association for Computational Linguistics, volume 1, pages 5579-5589.
>
> [3] Tianyi Zhang, Varsha Kishore, Felix Wu, Kilian Q. Weinberger, Yoav Artzi. 2019. Bertscore: Evaluating text generation with bert. arXiv preprint arXiv:1904.09675.

---

### Official Review · Reviewer_rB1x · 2023-08-12

**Soundness:** 3

**Excitement:**

4: Strong: This paper deepens the understanding of some phenomenon or lowers the barriers to an existing research direction.

**Paper Topic And Main Contributions:**

The authors proposed a generative boosting training approach for paraphrase identification, which uses seq2seq model to perform data augmentation on misclassified instances periodically. It shows that a single BERT model can outperform the state-of-the-art PI models with the boosting of the proposed approach. Experiments on the benchmarks of English QQP and Chinese LCQMC show that the method improves various PLM models.

**Reasons To Accept:**

1. The method seems promising in term of the efficiency, which is superior to the adversarial methods in this view.
2. The paper is well written and easy to follow.
3. All the source codes will be made publicly available to support reproducible research.

**Reasons To Reject:**

1. The soundness of mimicking the human learning process should be further discussed in detail.  What about multi-turn rethinking or correction for fine-grained errors？
2. More error ratio information and the corresponding examples should be analysed to support the motivation of the idea.

**Reproducibility:**

3: Could reproduce the results with some difficulty. The settings of parameters are underspecified or subjectively determined; the training/evaluation data are not widely available.

**Reviewer Confidence:**

3: Pretty sure, but there's a chance I missed something. Although I have a good feel for this area in general, I did not carefully check the paper's details, e.g., the math, experimental design, or novelty.

---

> ### Author Rebuttal · Authors · 2023-08-28
>
> We would like to express our heartfelt appreciation for dedicating your valuable time to reviewing our paper and offering us invaluable insights and suggestions. We have taken careful consideration of each of your suggestions, and we hereby provide our responses to each point as below.
>
>
>
> **----------Response to Comments----------**
>
>
> **Comment #1:** The soundness of mimicking the human learning process should be further discussed in detail. What about multi-turn rethinking or correction for fine-grained errors？
>
>
> **Response to #1:** We elaborate on our motivation in Lines 89-98 and subsequently validate our hypotheses through experimentation, wherein we set different boosting timings and ratios, as detailed in Section 4.5. The concept of multi-turn rethinking or correction as you suggested, is indeed an advisable approach to consider. We didn’t employ multi-turn correction indeed, due to the considerations of time efficiency.  In the future, we will investigate the method you have referenced, while ensuring efficiency as a foundational consideration.
>
>
> **Comment #2:** More error ratio information and the corresponding examples should be analysed to support the motivation of the idea.
>
>
> **Response to #2:** Due to the page limit, we didn’t include error rates and examples in the paper. We evaluated on the QQP verification set, using a Levenshtein similarity with a 0.6 threshold to determine literal similarity (sentences with a similarity greater than 0.6 were considered literally similar). Among the incorrect predictions made by the baseline model, we found that 35.9% were literally similar but not semantically similar (IBH0), and 14.7% were not literally similar but semantically similar (IBH1). These two categories constituted **50.7%** of all baseline model errors. In future revisions, we plan to provide a more detailed error analysis.
>
>
> We extend our heartfelt gratitude for your insightful comments, which will help us improve the quality of our paper.

---

### Meta-Review · Area_Chair_t5t2 · 2023-09-22

**Recommendation:** 4

**Metareview:**

As a reviewer indicates The work proposes a technique named Generative Boosting Training (GBT) where additional synthetic data is generated for difficult cases (where model makes mistake in prediction) on the task of paraphrase identification (PI). Authors propose to augment paraphrase sentences that are different in text but have similar semantic meaning (referred to as IBH1) and sentences that have similar text but different semantic meaning (referred to as IBH0). The work shows improvement from on 2 PI datasets, PPQ and LCQMC.


The main reasons to accept the paper are the following ones:

	The paper is well written and easy to follow.
	All the source codes will be made publicly available to support reproducible research.
   	The method seems promising in term of the efficiency, which is superior to the adversarial methods in this view.
    	 Good motivation to solve the paraphrase identification, with current limitations, detailed examples and vivid diagrams
    	Solid experiment on both BERT and GPT, reaching good performance or even state-of-the-art compared to the baseline models.
    	Explore the detailed of such data augmented boosting training, including but no limited to, boosting time, boosting ratio, different language models, etc.
    	GBT is efficient since it can be completed 1-hour on GPU.
	The GBT technique shows improvementand could be applied generally to other tasks as well. The negative and positive paraphrase augmentation seems like a good mix of balanced data augmentation (though we don’t know how much just adding IBH1 and just adding IBH0 independently contribute to improvement). The propose technique seems like a good choice for the PI task and some of the analyses (Figure 2 for boosting interval and ratio, table 5 for why to use GBT for hard examples) highlight interesting takeaways about GBT.


The main reasons to reject the paper are the following ones:

   	 The soundness of mimicking the human learning process should be further discussed in detail.
   	 More error ratio information and the corresponding examples should be analysed to support the motivation of the idea.
	The advanced data augmentation methods in comparison are from 2018-2020. It is unclear that whether the paper is comparing with the stat-of-the-art counterparts.
	It is not sure that whether part of the performance improvement can be attributed to the learned language ability from the seq2seq model via generated data (like the language model distillation), but not the GBT.
	 There are potentially numerous baselines as data augmentation for hard examples has several work. Given the closeness with this proposed work of using paraphrases (both negative and positive), some of baselines are necessary for comparison with GBT, especially counterfactual data-augmentation techniques as GBT uses (negative paraphrases in DA i.e., IBH0)
    	Some other, even more simpler baselines could be lower learning rate and training for more number of epochs. This would even strengthen the claims of GBT if improvements are significant.
   	 The ablation study in table 5 seemed more like good baselines which is good to have as it shows GBT is more effective when applied to only hard examples. A better ablation study could be giving just positive paraphrases (IBH1) and just negative paraphrases (IBH0)
    	Some of the important technical details are unclear. For example, which datasets and how was the paraphraser trained to generate candidate sentences for selecting IBH0 and IBH1. In line 268-277, more details would be needed as to how and where the 50K examples were selected from.
    	The text in line 293-295 makes the above point a little bit more unclear. It would be difficult for readers to understand and evaluate – “we manually observed the generated examples and find the results acceptable.”


Two reviewers are quite positive about the paper, one reviewer less, even after reading the answers provided by authors.

In sum,  despite the reasons to reject, the paper is well written, the results are reproducible, the techniques are interesting and the methods show improvements. Some clarifications should added if finally accepted.

---

### Decision · Program_Chairs · 2023-10-07

**Decision:**

Accept-Findings

**Comment:**

As a reviewer indicates The work proposes a technique named Generative Boosting Training (GBT) where additional synthetic data is generated for difficult cases (where model makes mistake in prediction) on the task of paraphrase identification (PI). Authors propose to augment paraphrase sentences that are different in text but have similar semantic meaning (referred to as IBH1) and sentences that have similar text but different semantic meaning (referred to as IBH0). The work shows improvement from on 2 PI datasets, PPQ and LCQMC.


The main reasons to accept the paper are the following ones:

	The paper is well written and easy to follow.
	All the source codes will be made publicly available to support reproducible research.
   	The method seems promising in term of the efficiency, which is superior to the adversarial methods in this view.
    	 Good motivation to solve the paraphrase identification, with current limitations, detailed examples and vivid diagrams
    	Solid experiment on both BERT and GPT, reaching good performance or even state-of-the-art compared to the baseline models.
    	Explore the detailed of such data augmented boosting training, including but no limited to, boosting time, boosting ratio, different language models, etc.
    	GBT is efficient since it can be completed 1-hour on GPU.
	The GBT technique shows improvementand could be applied generally to other tasks as well. The negative and positive paraphrase augmentation seems like a good mix of balanced data augmentation (though we don’t know how much just adding IBH1 and just adding IBH0 independently contribute to improvement). The propose technique seems like a good choice for the PI task and some of the analyses (Figure 2 for boosting interval and ratio, table 5 for why to use GBT for hard examples) highlight interesting takeaways about GBT.


The main reasons to reject the paper are the following ones:

   	 The soundness of mimicking the human learning process should be further discussed in detail.
   	 More error ratio information and the corresponding examples should be analysed to support the motivation of the idea.
	The advanced data augmentation methods in comparison are from 2018-2020. It is unclear that whether the paper is comparing with the stat-of-the-art counterparts.
	It is not sure that whether part of the performance improvement can be attributed to the learned language ability from the seq2seq model via generated data (like the language model distillation), but not the GBT.
	 There are potentially numerous baselines as data augmentation for hard examples has several work. Given the closeness with this proposed work of using paraphrases (both negative and positive), some of baselines are necessary for comparison with GBT, especially counterfactual data-augmentation techniques as GBT uses (negative paraphrases in DA i.e., IBH0)
    	Some other, even more simpler baselines could be lower learning rate and training for more number of epochs. This would even strengthen the claims of GBT if improvements are significant.
   	 The ablation study in table 5 seemed more like good baselines which is good to have as it shows GBT is more effective when applied to only hard examples. A better ablation study could be giving just positive paraphrases (IBH1) and just negative paraphrases (IBH0)
    	Some of the important technical details are unclear. For example, which datasets and how was the paraphraser trained to generate candidate sentences for selecting IBH0 and IBH1. In line 268-277, more details would be needed as to how and where the 50K examples were selected from.
    	The text in line 293-295 makes the above point a little bit more unclear. It would be difficult for readers to understand and evaluate – “we manually observed the generated examples and find the results acceptable.”


Two reviewers are quite positive about the paper, one reviewer less, even after reading the answers provided by authors.

In sum,  despite the reasons to reject, the paper is well written, the results are reproducible, the techniques are interesting and the methods show improvements. Some clarifications should added if finally accepted.